# Ovotransferrin Supplementation Improves the Iron Absorption: An In Vitro Gastro-Intestinal Model

**DOI:** 10.3390/biomedicines9111543

**Published:** 2021-10-26

**Authors:** Rebecca Galla, Paride Grisenti, Mahitab Farghali, Laura Saccuman, Patrizia Ferraboschi, Francesca Uberti

**Affiliations:** 1Laboratory Physiology, Department of Translational Medicine, University of Eastern Piedmont, Via Solaroli 17, 28100 Novara, Italy; rebecca.galla@uniupo.it (R.G.); mahitab.farghali@uniupo.it (M.F.); laura.saccuman@uniupo.it (L.S.); 2Bioseutica B.V., Landbouwweg 83, 3899 BD Zeewolde, The Netherlands; pgrisenti@bioseutica.com; 3Department of Medical Biotechnology and Translational Medicine, Università degli Studi di Milano, Via Saldini 50, 20133 Milan, Italy; patrizia.ferraboschi@unimi.it

**Keywords:** ovotransferrin, bovine lactoferrin, iron metabolism, gastro-intestinal barrier, iron absorption, iron transportation

## Abstract

Transferrins constitute the most important iron regulation system in vertebrates and some invertebrates. Soluble transferrins, such as bovine lactoferrin and hen egg white ovotransferrin, are glycoproteins with a very similar structure with lobes that complex with iron. In this in vitro study, a comparison of bovine lactoferrin and ovotransferrin was undertaken to confirm the comparability of biological effects. An in vitro gastric barrier model using gastric epithelial cells GTL-16 and an in vitro intestinal barrier model using CaCo-2 cells was employed to evaluate iron absorption and barrier integrity. An analysis of the molecular pathways involving DMT-1 (divalent metal transporter 1), ferritin and ferroportin was also carried out. These in vitro data demonstrate the activity of both 15% saturated and 100% saturated ovotransferrin on the iron regulation system. Compared with the commercial bovine lactoferrin, both 15% saturated and 100% saturated ovotransferrin were found to act in a more physiological manner. Based on these data, it is possible to hypothesise that ovotransferrin may be an excellent candidate for iron supplementation in humans; in particular, 15% saturated ovotransferrin is the overall best performing product. In vivo studies should be performed to confirm this in vitro data.

## 1. Introduction

Iron is an essential element that plays a central role in many metabolic processes. It is necessary for most, if not all, pathways involved in energy and substrate metabolism. Iron plays a major role in (1) oxygen transport (haemoglobin) and short-term oxygen storage (myoglobin); (2) haem enzymes involved in electron transfer (e.g., cytochromes a, b and c, and cytochrome c oxidase) and oxidase activities (e.g., cytochrome P-450 mixed function oxidases, oxidases and peroxidases); and (3) iron–sulphur clusters in energy transduction and oxido-reductase activities (e.g., succinate, isocitrate and NADPH dehydrogenase, xanthine oxidases). It is also a cofactor in various non-haem-containing enzymes (e.g., phenylalanine, tryptophan and tyrosine hydroxylases, and proline and lysine hydroxylases) [1]. Iron, naturally present in foods and beverages, is inefficiently and variably absorbed into the body depending on dietary and host-related factors. Iron absorption occurs mainly in the duodenum and proximal small intestine, and involves the uptake from the lumen into enterocytes. It is not possible to predict the exact percentage of iron that is absorbed from the diet; it is suggested that approximately 10–20% of an oral iron dose is absorbed [2]. A portion of the absorbed fraction is then translocated out of the enterocytes across the basolateral membrane by a transmembrane basal transporter (ferroportin 1) to carriers in the plasma of the portal circulation. Apotransferrin, a glycoprotein present in biological fluids, binds one or two ferric iron ions and transfers them to the liver and systemic circulation. The liver is the main iron depot of the body where iron is stored as the soluble protein complex ferritin, a globular protein present in every cell type and, to a lesser extent, as ferritin-derived insoluble haemosiderin [3]. Iron deficiency is common, especially among women in the fertile period (due to menstrual losses, pregnancy and breastfeeding) and in postmenopausal women [4] or in subjects who have undergone major surgery, physical trauma, having gastrointestinal diseases (e.g., celiac disease, inflammatory bowel disease, ulcerative colitis, Crohn’s disease and peptic ulcer disease) and in people who have a diet that is low in iron, such as vegans and vegetarians whose diets do not include iron-rich foods (it is well-known that haem iron is better absorbed then non-haem iron). Iron-deficiency (ID) is defined as the decrease in the total content of iron in the body and is traditionally classified by severity: the mildest form has a reduced (“inadequate”) iron store but no evidence of impaired iron delivery to the functional compartment (low serum ferritin (SF)); the mild form, also called “iron-deficient erythropoiesis”, is characterised by a disparity between the rate of delivery of absorbed iron and iron released from the stores and the cellular requirements (reduced serum iron (SI)), increased total iron binding capacity (TIBC), reduced percentage saturation of transferrin (% Sat), increased plasma transferrin receptor concentration (TfR) and increased red blood cell zinc protoporphyrin. The most severe form, called Iron Deficiency Anaemia (IDA), where there is evidence of a deficiency in the major functional compartment, the circulating red blood cell mass, is established based on a functional iron deficiency: iron-deficient red blood cells reveal evidence of inadequate haemoglobin synthesis and haemoglobin content [5]. It is important to ensure a prompt treatment in patients with ID and IDA because proper management may improve quality of life, alleviate the symptoms of iron deficiency (such as fatigue, paleness, shortness of breath, headaches, and heart palpitation) and possible cognitive deficits, and reduce the need for blood transfusions. The amount of iron supplementation recommended to treat iron deficiency anaemia for adults is 120 mg/day iron for 3 months. For infants and younger children, it is 3 mg/kg/day, not to exceed 60 mg daily [6].

A recent review of pregnant women with IDA showed that the daily oral administration of bovine lactoferrin (bLact) is as effective as ferrous sulphate in improving haematological parameters (increasing Hb, serum ferritin, and iron levels) with fewer gastrointestinal side effects [7]. bLact is a glycoprotein from the transferrin family present at high concentrations in the milk of humans and other mammals. This molecule has an isoelectric point of 8.1, has twice the affinity for iron than human serum transferrin, and is able to reversibly chelate two Fe^3+^ ions per molecule. Hashim et al. [7] showed that the administration of 100 mg of bovine lactoferrin twice a day before meals for four weeks is equivalent to the administration of 100 mg of elemental iron. CN1850270A discloses the use of another iron-binding glycoprotein from the hen’s egg belonging to the family of transferrin, namely, ovotransferrin (holo-OvT) saturated with trivalent iron ions as iron supplement/fortification, wherein the molar ratio of ovotransferrin and iron ions in the complex is 1:0.5–2, which corresponds to a degree of iron saturation of OvT ranging from 100% to 25%. This degree of iron saturation of hen egg ovotransferrin tested on an animal model (Kunming mice) showed a potential therapeutic use for treating iron deficiency.

OvT (previously also known as conalbumin) is an egg white glycoprotein responsible for the transfer of ferric ions from the hen oviduct to the developing embryo. This protein is made up of a single polypeptide chain of 686 amino acids with a molecular mass of about 78–80 kDa and a single glycan chain (composed of mannose and *N*-acetylglucosamine residues) in the C-terminal domain [8], and shows an isoelectric point of 6.0. OvT is synthesised in the hen oviduct and accumulated in the albumen fraction of eggs where it represents about 12–13% of total egg white proteins. OvT contributes to promoting the growth and development of the chicken embryo, mainly by preventing the growth of micro-organisms together with other proteins such as lysozyme [9], cystatin [10], ovomacroglobulin [11] and avidin [12]. Similarly to other transferrins, OvT is a two-lobed protein possessing the capability to reversibly bind two Fe^3+^ ions per molecule, along with two CO_3_^2−^ or HCO^3−^ ions with high affinity, thus delivering iron into host cells by membrane-bound specific receptors [13]. The potential use of OvT as a nutritional ingredient has also been reported by several studies [14,15,16]. OvT appears in two forms: the apo and holo-form. The apo-OvT form does not contain iron whereas the holo-form may contain up to about 1.4 mg of iron/gram of protein (100% saturated). The holo-OvT form seems more stable against proteolytic hydrolysis and heat denaturation than the apo-OvT form [17]. In November 2020, OvT (unspecified if holo or apo type) was used in a clinical trial to evaluate the efficacy and safety of OvT in COVID-19 patients as an immunomodulator in addition to the standard of care therapy, and this has been included in a Clinical Trial Registry [18].

The potential of OvT as a dietary food supplement in ID or IDA seems most likely to be associated with the holo-OvT form in which the degree of saturation ranges from 100% (two moles of iron per molecule of OvT) and 25% (0.5 mole of iron per molecule of OvT), but biochemical and clinical evidence of its efficacy are still lacking, in addition to evidence on the efficacy of the apo-OvT form or OvT having a very low degree of iron saturation. Moreover, the isoelectric point of OvT (6.0) suggests that in acidic environments, like in vivo gastric juices, the stability of the holo-OvT form can be modified releasing “one pot” of the iron salts in the stomach, which can potentially lead to side effects usually associated in iron supplementation with inorganic and organic iron salts [19,20].

To try to clarify these as-yet unexplored aspects, this study was performed to assess the in vitro release and absorption of iron from OvT, in addition to the gastro-intestinal cell tolerability.

## 2. Materials and Methods

### 2.1. Reagent Preparation

Purified ovotransferrin (OvT) from hen egg white was supplied by BIOSEUTICA BV (Zeewolde, NL) with different degrees of iron saturation: apo-form (extremely low iron content, 5.8 ppm iron, named apo-OvT) and holo-form (iron-bound) at 15% (named holo-OvT 15%) (152 ppm iron) and 100% saturation (named holo-OvT 100%) (1.400 ppm iron). The preparations of these substances are related to the European patent application N. EP21158368.7 by BIOSEUTICA BV, The Netherlands.

Bovine Lactoferrin (bLact) was purchased from the market (Biocon, LTD, Aichi, Japan), analysed by ICP (iron content 157 ppm, corresponding to around 11% iron saturation).

In the first set of experiments, a dose response study of apo-OvT, holo-OvT 15%, holo-OvT 100% and bLact was carried out ranging from 30 to 200 μg/mL as reported in literature [21]. Then two concentrations were maintained for all successive experiments: 50 and 30 μg/mL for both OvT forms and bLact. The substances were dissolved directly in the stimulation medium, Dulbecco’s Modified Eagle Medium (DMEM) without red phenol (Merck Life Science, Rome, Italy) and foetal bovine serum (FBS, Merck Life Science, Milan, Italy) but supplemented with 1% penicillin/streptomycin (Merck Life Science, Rome, Italy), 2 mM L-glutamine (Merck Life Science, Rome, Italy) and 1 mM sodium pyruvate (Merck Life Science, Italy) to make a 20× concentration, and then diluted in the same medium to obtain 1×.

### 2.2. Cell Culture

The Caco-2 cell line, supplied by the American Type Culture Collection (ATCC), was cultured in Dulbecco’s Modified Eagle’s Medium/Nutrient F-12 Ham (DMEM-F12, Merck Life Science, Rome, Italy) containing 10% FBS, 2 mM L-glutamine and 1% penicillin–streptomycin at 37 °C in an incubator at 5% CO_2_ [22]. This cell line is used to carry out a widely accepted (by EMA and FDA) experimental model to predict absorption, metabolism and bioavailability of drugs and xenobiotics following oral intake [23,24,25,26,27]. The cells were used at passage numbers between 26 to 32 to preserve the physiological balance between paracellular permeability and transport properties [22]. Cells were cultured in different way based on different experimental protocols: 1 × 10^4^ cells in 96 well plates to study cell viability by MTT-based In Vitro Toxicology Assay Kit (Merck Life Science, Rome, Italy); 1 × 10^6^ cells in 6 well plates to study the intracellular mechanisms involved by Western blot analysis; 2 × 10^4^ cells on 6.5 mm Transwell^®^ with 0.4 μm pore polycarbonate membrane insert (Merck Life Science, Italy) in a 24 well plate to perform an absorption study, and 1.4 × 10^6^ to prepare a 3D system (VITVO^®^, Rigenerand, Medolla, Italy). Before the stimulation, cells were washed and incubated for 8 h in DMEM without red phenol and supplemented with 0.5% FBS, 2 mM L-glutamine and 1% penicillin-streptomycin at 37 °C. Cells plated on Transwell^®^ insert were maintained in complete medium changed every other day, first basolaterally and then apically for 21 days before the stimulations [28]. The treatment was evaluated over a period of 6 h (the maximum time of absorption of the molecules) at the time points of 1, 2, 3, 4, 5 and 6 h. The same conditions have been reproduced in the presence of 50 μM Fe^3+^ [29].

The GTL-16 cell line, donated by the Laboratory of Histology of the University of Eastern Piedmont (Italy), is a clonal line derived from a poorly differentiated gastric carcinoma cell line [30] widely used as a model of gastric epithelial cells. Cells were cultured in Dulbecco’s Modified Eagle Medium (DMEM, Merck Life Science, Rome, Italy) supplemented with 10% FBS, 1% penicillin-streptomycin in incubator at 37 °C, 5% CO_2_ [28,31]. Cells were cultured in a different way based on different experimental protocols: 1 × 10^4^ cells to explore the cell viability by MTT; on 60 mm dishes until confluence to analyse the intracellular pathways by Western blot; 2 ×10^4^ into 6.5 mm translucent polyethylene terephthalate (PET) Transwell^®^ insert 0.4 μm (Greiner bio-one, Kremsmünster, Austria) in a 24 well plate to study absorption; and 5.6 × 10^5^ to create a 3D system (VITVO^®^).

Before stimulations, cells were incubated with DMEM without red phenol and FBS but supplemented with 1% penicillin/streptomycin, 2 mM L-glutamine and 1 mM sodium pyruvate at 37 °C in an incubator overnight to synchronise [28]

The cells plated on Transwell^®^ insert were maintained in complete medium changed every other day, first basolaterally and then apically for 7 days before the stimulations. The stimulations were evaluated within 4 h (the maximum absorption time of the molecules) at the points as follows: 1, 2, 3 and 4 h. The same conditions have been reproduced in presence of 50 μM Fe^3+^ to assess the mechanism similar to that occurring during food supplementation [29].

### 2.3. In Vitro Gastric Barrier Model

During the 7 days after seeding GTL16 cells on the apical side of 24 well Transwell^®^ permeable support, the transepithelial electrical resistance (TEER) was continuously measured by an EVOM3 Voltohmmeter (World Precision Instruments, Sarasota, FL, USA) and the experiments were started when TERR reached ≥150 Ω·cm^2^ [32]. In particular, to study the effects on absorption from apical-to-basolateral pH gradients, after 7 days the medium was changed on both the apical and basolateral sides adding HCl (Merck Life Science, Italy) to the medium to obtain pH 3 at the apical side for 60 min, as reported in the literature [28,33]. The stimulations were performed in the same manner and conditions as previously described, and then iron and transferrin quantifications were measured by ELISA assays as reported above.

### 2.4. In Vitro Intestinal Barrier Model

The TEER values of the inserts were measured on the alternate days continuously for 21 days using EVOM3 and the experiments were started when TERR reached ≥400 Ω·cm^2^ [34]. In literature it is reported that the TEER values ≥260 ± 65Ω·cm^2^ are recommended for the transport study [35].

After 21 days, two different concentrations of OvT and bLact were added to the apical culture medium which has pH 6.5 (mimicking the acid condition of lumen of the small intestine), whereas the basolateral has 7.4 (neutral pH like the blood), as reported in the literature [28]. During treatments the cells were maintained in an incubator at 5% CO_2_, and at the end of stimulations the iron and transferrin quantifications were measured by ELISA.

### 2.5. D Gastro-Intestinal Tract In Vitro Model

To evaluate if the two models, gastric and intestinal, separately observed, can reproduce the human condition of increasing the plasma iron, a further three-dimensional (3D) model was used that allows direct communication between the two cell populations (gastric and intestinal): the 3D model VITVO^®^ bioreactors were prepared following the manufacturer’s instruction [36]. Briefly, the model was first primed with 1.4 mL of media alone by using a 2.5 mL syringe (Becton Dickinson and Co, Franklin Lakes, NJ, USA) to ensure a complete wetting of the 3D matrix. After 1 h, GTL-16 cells were inserted at 5.6 × 10^5^ with 1.4 mL of DMEM 10% FBS using a 5 mL syringe that was connected with the inlet port; in addition, the closing cap of the outlet port was removed in order to remove the media without cells used to hydrate VITVO^®^. Once the loading was complete, the outlet port was closed, the syringe removed, and the inlet port was closed; the bioreactor was put in an incubator with the loading side face up. The cells were maintained in culture for one week, changing the medium every 24 h.

The same protocol was performed for the intestinal cell line, Caco-2. Briefly, after hydrating the matrix of 3D model VITVO^®^ bioreactors, Caco-2 cells were injected at 1.4 × 10^6^ with a 5 mL syringe in the bioreactor with DMEM/F12 as previously described and put in an incubator at 37 °C and 5% CO_2_ for two weeks, changing the medium every 24 h. This second bioreactor was connected to the first, which contained GTL-16 cells in order to simulate the transit of a molecule through the stomach and subsequently into the intestine.

Before stimulations, each bioreactor was checked at microscopy to verify the confluence and the medium was changed to reproduce the condition present in the specific environment; adding HCl to the medium (DMEM without red phenol and FBS but supplemented with 1% penicillin/streptomycin, 2 mM L-glutamine and 1 mM sodium pyruvate) to obtain pH 3 to the GTL-16 cell and pH 6.5 to mimic the acid condition of lumen of the small intestine on Caco-2. Two fundamental parameters were analysed, namely, the concentration of transferrin and iron.

### 2.6. MTT Assay

An MTT test was performed following a standard protocol [28]. Briefly, after stimulations, both cell types were incubated with 1% of MTT dye in DMEM white for 2 h at 37 °C in an incubator, and then purple formazan crystals were dissolved in an equal volume of MTT solubilisation solution. Cell viability was determined by measuring the absorbance at 570 nm with correction at 690 nm, through a spectrometer (VICTOR × 4 Multilabel Plate Reader, PerkinElmer, Waltham, MA, USA), and calculated by comparing results to control cells which were maintained for the same time in the same medium and conditions used for treatment samples, but without any stimulus (baseline 0%).

### 2.7. Iron Quantification Assay

The quantification of total iron in GTL-16 and Coco-2 cells was performed using an Assay Kit (Merck Life Science, Rome, Italy) able to quantify ferrous iron (Fe^2+^), ferric iron (Fe^3+^) and total iron (total iron—ferrous iron) following the manufacturer’s instructions. The measurements were carried out in both apical and basolateral environments adding 5 μL of Iron Reducer to each of the sample wells to reduce Fe^3+^ to Fe^2+^ and incubating samples for 30 min at room temperature (RT), protected from light. After that, 100 μL of Iron Probe was added to each well containing standard or test samples and incubated for 60 min at RT in the dark and the absorbance at 593 nm (A593) was measured by a spectrometer (VICTOR × 4 Multilabel Plate Reader; results were generated by a standard curve and are expressed as ng/μL [28].

### 2.8. Transferrin Quantification Assay

The Human Transferrin ELISA kit (Thermo Fischer, Milano, Italy) is designed to measure the amount of the target bound between a matched antibody pair following the manufacturer’s instructions. Briefly, 100 μL of diluted samples were incubated overnight at 4 °C, washed 4 times with 1× Wash Buffer, and then 100 μL of biotin conjugate was added to each well and incubated for 1 h at room temperature with gentle shaking. Then the wells were washed 4 times and 100 μL of streptavidin-HRP and 100 μL of TMB Substrate were added. Finally, the plate was incubated for 30 min at room temperature in the dark with gentle shaking and the reaction stopped with 50 μL of Stop Solution. The absorbance was measured by spectrometer at 450 nm (VICTOR × 4, multilabel plate reader) and calculated by comparing results to control cells (baseline 0%).

### 2.9. SOD Assay

The level of SOD was measured following the manufacturer’s instructions (Cayman’s Superoxide Dismutase Assay Kit) which reveals all three types of SOD (Cu/Zn, Mn, and FeSOD). Briefly, in a 96 well plate, the level of SOD present on 3D cell lysates was measured by comparing data to a standard curve (0.05–0.005 U/mL). The absorbance of all samples was measured through a spectrometer (VICTOR × 4 multilabel plate reader) at 480 nm and the results are expressed as a means (%) compared to control [37].

### 2.10. Western Blot

At the end of 3D stimulation, VITVO^®^ was washed with cold 1.4 mL of Phosphate Buffered Saline (PBS) 1× supplemented with 1:50 mix Phosphatase Inhibitor Cocktail (Merck Life Science, Rome, Italy) and 1:200 mix Protease Inhibitor Cocktail (Calbiochem, San Diego, CA, USA) to completely remove the remaining culture medium. Subsequently, the matrix was extracted from VITVO^®^ using a surgical scalpel in order to cut carefully the transparent oxygenation membranes. The VITVO^®^ matrix was transferred to a Petri dish, in which it was cut into smaller pieces with scissors and these were lysed in ice with cold Complete Tablet Buffer (Roche, Milan, Italy) supplemented with 2 mM sodium orthovanadate, 1 mM phenylmethanesulfonyl fluoride (PMSF; Merck Life Science, Rome, Italy), 1:50 mix Phosphatase Inhibitor Cocktail (Merck Life Science, Italy) and 1:200 mix Protease Inhibitor Cocktail (Calbiochem, San Diego, CA, USA). The pieces of matrix were also lysed with a scraper so as to homogenise the cellular component with the lysis buffer, and subsequently the whole was resuspended with a tip and transferred to an Eppendorf tube. Eppendorf tubes were centrifuged at 13,500 rpm at 4 °C for 30 min, and then the supernatants were divided from pellet. A quantity of 40 μg of proteins of each sample was resolved on 5% and 15% SDS-PAGE gels and the PVDF membranes (Polyvinylidene difluoride membrane, Thermo Fischer) were incubated overnight at 4 °C with specific primary antibodies, such as anti-ferritin 1:1000 (Santa-Cruz, CA, USA), anti-ferroportin 1:2000 (Santa-Cruz, CA, USA) and anti-DMT-1 1:400 (Santa-Cruz, CA, USA). The protein expression was normalised and verified through β-actin detection (1:5000; Merck Life Science, Rome, Italy) and expressed as mean SD (% vs. control).

### 2.11. Occludin Quantification Assay

The Human Occludin (OCLN) ELISA kit (MyBiosource, San Diego, CA, USA) was used to measure the presence of occludin in Caco-2 lysates following the manufacturer’s instructions. Briefly, the cells were lysed with cold PBS 1× and centrifuged at 1500× *g* for 10 min at 4 °C. A quantity 100 μL of each sample was added to a strip well and incubated at 37 °C for 90 min; then, the supernatants were removed and 100 μL of Detection Solution A was added to each of them and incubated for 45 min at 37 °C. After this time, the wells were washed with Wash Solution and incubated with 100 μL of Detection Solution B for 45 min, and then 90 μL of Substrate Solution was added, incubating for 20 min at 37 °C in the dark. Finally, after adding 50 μL of Stop Solution, the plate was analysed by a spectrometer (VICTOR × 4, multilabel plate reader) at 450 nm. The concentration is expressed as ng/mL compared to a standard curve.

### 2.12. Claudin 1 Detection Assay

The Human Claudin1 ELISA kit (MyBiosource, San Diego, CA, USA) was measured in Caco2 lysates, following the manufacturer’s instructions. Briefly, the cells were lysed with cold PBS 1× and centrifuged at 1500× *g* for 10 min at 4 °C. A quantity of 100 μL of each sample was added to a well and incubated at 37 °C for 90 min; then, the materials were removed and, to each well was added 100 μL of Detection Solution A, and incubated for 45 min at 37 °C. After this time the wells were washed and 100 μL of Detection Solution B in each well was added and then incubated for 45 min. Then, 90 μL of Substrate Solution in each well was also added and the plate incubated for 20 min at 37 °C in the dark. A quantity of 50 μL of Stop Solution was used to stop the reaction and the plate was analysed by a spectrometer (VICTOR × 4, multilabel plate reader) at 450 nm. The concentration is expressed as ng/mL comparing data to standard curve.

### 2.13. Human Tight Junction Protein 1 (ZO-1) Detection

The Human Tight Junction Protein 1 (TJP1) ELISA kit (MyBiosource, San Diego, CA, USA) was used to measure the presence of tight junction protein 1 or Zona occludens 1 (ZO1) in cell lysates in Caco-2, following the manufacturer’s instructions. Briefly, the cells were lysed using trypsin and then collected by centrifugation. Then, cells were washed three times with cold PBS 1× and then resuspended in PBS 1×; subsequently, the cells were subjected to ultrasonication four times and then they were centrifuged at 1500× *g* for 10 min at 4 °C in order to remove cellular debris. A quantity of 100 μL of each sample was added to a well and incubated at 37 °C for 90 min; then, the materials were removed, and to each well was added 100 μL of Detection Solution A and incubated for 45 min at 37 °C. Wells were washed with Wash Solution and after 100 μL Detection Solution B was added to each well. After incubation of 45 min, the wells were washed again and 90 μL of Substrate Solution was added in each well and then incubated for 20 min at 37 °C in the dark. Finally, 50 μL of Stop Solution was added and then the plates were read by a spectrometer (VICTOR × 4, multilabel plate reader) at 450 nm. The concentration is expressed as ng/mL.

### 2.14. Statistical Analysis

Data obtained from each experimental protocol reproduced in at least five independent experiments performed in quadruplicate were collected and analysed by Prism GraphPad statistical software. One-way analysis of variance (ANOVA), followed by Bonferroni post hoc tests, were carried out for statistical analysis to compare groups and pairwise differences compared by Mann–Whitney U tests. All results are expressed as means ± SD and differences are statistically significant with a *p*-value < 0.05.

## 3. Results

### 3.1. Dose-Response and Time-Dependent Study on GTL-16 and Caco-2 Cells

Different concentrations in a range from 30 to 200 μg/mL of apo-OvT and holo-OvT forms (15% and 100% iron saturation) and bLact were tested for 24 h to identify the concentration able to induce a greater effect on the viability of GTL-16 and Caco-2 cells before starting the experiments. The selected concentrations for all the tested products were: 200, 100, 50 and 30 μg/mL. Compared to the control, all the OvT samples tested showed a positive effect on cell viability of GTL-16 cells (Figure 1). Based on the results, the two lower concentrations of 50 and 30 μg/mL of the three OvTs tested showed an increase in cell viability compared to the control (about *p* < 0.05) and compared to the higher concentrations (*p* < 0.05), and the lower dosage had a similar or superior profile to that of bLact at the same concentrations (holo-OvT 100% *p* < 0.05). All these data confirmed that the stimulations acted without cytotoxicity on the gastric cells and holo-OvT forms appeared to have a better effect than apo-OvT; however, all OvT forms appeared to have the best performance in terms of mitochondrial metabolic well-being compared to bLact.

Because the absorption of the molecules occurs mainly within the intestinal tract, additional experiments were carried out in the similar conditions on Caco-2 cells. As shown in Figure 2, the lower concentrations of holo-OvT of 15% and 100% saturated were confirmed to have the greater effects on cell viability compared to the control (*p* < 0.05) and to the other concentrations; in addition, 50 μg/mL holo-OvT 15% was able to induce the highest increase In mitochondrial performance (*p* < 0.05) compared to all other OvT forms and to bLact (*p* < 0.05 for all concentrations). The results showed that OvT is able to act similarly to bLact when used in specific forms and dosages. Also in this context, no cytotoxicity effects were observed on intestinal cells. Taken together, these preliminary data support the hypothesis that OvT appeared to have the best performance in terms of mitochondrial metabolic well-being, and two concentrations, 30 and 50 μg/mL, seemed to be a better choice. These findings are similar to those observed in the bLact effects at the same concentrations in both cell types. For this reason, these two concentrations were used for all successive experiments.

In order to confirm the data reported above, GTL-16 and CaCo2 cells were stimulated with holo-OvT 100%, holo-OvT 15%, apo-OvT, and bLact at the selected concentrations (50 and 30 μg/mL) for 1–3 h on gastric cells and for 1–6 h on the intestinal compartment. At the end of the stimulations, an MTT test was performed to evaluate cell viability. As shown in Figure 3, a time-dependent effect was revealed on GTL-16 cells with a maximum effect of holo-OvT 100% and 15% saturated at 3 h (Figure 3c) compared to the control (*p* < 0.05) and to other time-points (*p* < 0.05). The concentration of 30 μg/mL holo-OvT 15% was shown to be the best compared to the control throughout the entire analysis process at different timeframes, and when compared to 30 μg/mL holo-OvT 100% (about 30.7%, *p* < 0.05) or to 30 μg/mL bLact (about 98%, *p* < 0.05). Similar findings were also observed at 3 h with 50 μg/mL holo-OvT 15% compared to 50 μg/mL holo-OvT 100% (about 2.5 times, *p* < 0.05) or to 50 μg/mL bLact (*p* < 0.05). In addition, both 50 and 30 μg/mL holo-OvT 15% were also able to maintain the effect at 24 h. indicating a slow-release mechanism with a better physiologic profile (*p* < 0.05) compared to 50 and to 30 holo-OvT 100% (about 120% times and about 22%, respectively, *p* < 0.05), as illustrated above in Figure 1. Finally, the apo-OvT at both concentrations had no significant effect on cell viability, supporting the hypothesis of the iron saturation rate to obtain a beneficial effect of ovotransferrin.

Regarding the CaCo-2 cells, a time-dependent effect (Figure 4) was also observed for all OvT forms and bLact at both 30 and 50 μg/mL; in particular, 30 μg/mL holo-OvT 15% was able to induce a greater effects compared to the control (*p* < 0.05) and compared to the same OvT form at 50 μg/mL (*p* < 0.05) and to the other molecules (*p* < 0.05) during all period analysed. The main result was observed at 3 h (Figure 4b) of stimulation (about 39%) and then decreased slowly until 24 h (about 16%, as illustrated in Figure 2). These results obtained in both cell types confirmed that OvT has a time-dependent effect and suggest that 30 μg/mL holo-OvT 15% may be the best option compared to the control (*p* < 0.05) and also compared to the bLact tested (*p* < 0.05), which is more evident at 3 h (about 52% compared to 30 μg/mL bLact and about 44% compared to 50 μg/mL bLact). Furthermore, based on the results, one dosage in 24 h in humans can be assumed. In addition, because the apo-OvT induced weaker results compared to holo-OvT forms, probably due to the lack of iron binding, this product was not used in the absence of iron-pre-treatment experiments.

### 3.2. Analysis of Transferrin after Gastric and Intestinal Barrier In Vitro

Because the main important finding about the use of lactoferrin is to modulate the iron absorption in humans, it is more important to verify the passage of all OvT forms compared to bLact mimicking the human oral intake. In this context, some experiments were performed to evaluate the absorption of the tested products measuring transferrin on both GTL-16 and CaCo-2 cells plated on a Transwell^®^ system over time (ranging from 1 to 6 h) mimicking the gastro-intestinal digestion time in humans. As shown in Figure 5, the analysis of the basolateral environment of the gastric compartment (GTL-16 cell) showed a slow but constant gastric absorption over time (1–3 h), confirming again a “slow-release effect” of holo-OvT 15%, substantiating its best iron saturation level of both concentrations tested compared to bLact (*p* < 0.05) or holo-Ovt 100% (*p* < 0.05). This is important data to define both the type of dose form (e.g., slow or fast release tablets) and the posology in humans to prevent the rebound effect.

However, because the important data which can explain the real absorption rate of all the tested products relate to the intestinal barrier, additional tests were carried out on Caco-2 cells to analyse the basolateral environment (Figure 6). The analysis with the ELISA method showed a slow but constant absorption over time (from 1 to 6 h), confirming a “slow-absorption effect” following a physiological trend with a maximum effect at 3 h leading to a reduction over time for all OvT forms and bLact. In particular, the data showed that 30 μg/mL holo-OvT 15% had a comparable effect to 30 μg/mL bLact (about 92%) and a better effect compared to the higher concentration of 50 μg/mL of bLact (about 21%), confirming the importance of the dosage in addition to the saturation rate.

Based on these data, holo-OvT 15% at the concentration of 30 μg/mL was the best concentration compared to the other products tested.

### 3.3. Three-Dimensional (3D) In Vitro Model Mimicking In Vivo Complexity of the Gastro-Intestinal Barrier

To obtain further information on the physiological absorption and bioavailability in an integrated gastro-intestinal model, a further experiment using a three-dimensional (3D) validated in vitro model was performed in order to mimic the in vivo complexity of the gastro-intestinal barrier. An intestinal barrier was recreated in vitro connecting the basolateral part from the gastric barrier directly to the apical of intestinal barrier and, subsequently, the basolateral part was collected and analysed to evaluate the quantity passed through the barrier. This model puts the two cell populations in direct communication, as in a living organism, and for this reason it is considered an optimal pre-clinical model. In this context, 50 and 30 μg/mL of holo-OvT 100% and 15 were tested in the same timeframe previously used (1, 3 and 6 h). The data obtained from this 3D model (Figure 7) confirmed that the gastro-intestinal adsorption has a physiological trend. In particular, it was observed that the predigested product is absorbed with a kinetic having a maximum effect at 3 h followed by a slow decrease at 6 h that does not end, indicating a kind of "slow absorption" that can suggest an in vivo single administration in 24 h. From this test, the best performance and bioavailable concentration was induced by 30 μg/mL of holo-OvT 15% compared to the control (*p* < 0.05). This was further confirmation that this holo-OvT 15% is more bioavailable compared to bLact at the same concentration (*p* < 0.05, about 21%) and comparable to 50 μg/mL bLact (*p* < 0.05, about 92%) at 3 h of stimulation. These results demonstrated that 30 μg/mL holo-OvT 15 is more bioavailable than 100% holo-OvT at the same concentration. This is most evident at 3 h (approximately 88%). holo-OvT 15% also appears to be better than bLact, which exhibits rapid absorption but lower bioavailability.

Finally, to exclude any stress caused by stimulations in a complex system such as the human gastrointestinal system, the analysis of SOD activity was also performed in the basolateral environment of the intestinal compartment after gastric digestion at the same time reported above. As reported in Figure 8, holo-OvT 100% and 15% at both concentrations were able to maintain the level of SOD activation near control values and the activation was less compared to bLact during all period analysed. These results supported the degree of safety and the lack of negative effects or irritative effects on the lumen of the intestinal epithelium, which is more important to maintain the integrity of tight junctions (TJ). Between the two OvT forms, the holo-OvT 15% confirmed its beneficial effects compared to the 100% saturated form, and this effect was observed during all analysed periods (*p* < 0.05) and in the presence of both 50 and 30 μg/mL. In addition, comparing 50 and 30 μg/mL of holo-OvT 15%, the best performance was obtained with 30 μg/mL during all periods analysed because no significant changes were observed. Finally, compared to bLact, the most evident effect was observed at 3 h in which 30 μg/mL holo-OvT 15% caused a main reduction compared to 30 μg/mL bLact (about 90%) and to 50 μg/mL (about 94%), thus confirming it to be a better choice with respect with bLact.

### 3.4. Analysis of the Mechanism of Action by Intracellular Mechanism Activated on 3D Model

To better define the mechanisms involved in both gastric and intestinal environments, different markers such as DMT1, ferritin, and ferroportin were evaluated by Western blot and densitometric analysis on a 3D system treated with 50 and 30 μg/mL of all OvT forms and bLact. As reported in Figure 9, DMT-1 was expressed in both GTL-16 and CaCo-2 cells (Figure 9a,b, respectively), demonstrating that OvT has the same iron transport mechanism as bLact. In particular, the effects on DMT-1 expression were greater with holo-OvT 15% in GTL-16 cells, which was more evident with the 30 μg/mL concentration compared to the control (*p* < 0.05) and to 50 μg/mL (about 2.2%), and to both concentrations of holo-OvT 100% (about 42% to 30 μg/mL and about 51% to 50 μg/mL *p* < 0.05), and even to the two concentrations of bLact tested (about 85% to 30 μg/mL and about 13.2% to 50 μg/mL, *p* < 0.05). Regarding Caco-2 cells (Figure 9b), holo-OvT 15% at 30 μg/mL showed a higher expression of DMT-1 compared to the 50 μg/mL concentration (about 12%) and to both concentrations of bLact (about 45% to 30 μg/mL and about 30% to 50 μg/mL *p* < 0.05) and also to holo-OvT 100% at the same concentration (about 31%, *p* < 0.05). These findings supported the effects observed previously about the transferrin quantification, indicating a better effect of holo-OvT 15% compared to all other products tested.

With the aim to better understand the biochemistry of the tested products on iron metabolism, the effects on the different iron transport chains were evaluated. In particular, the transmembrane protein ferroportin and the intracellular protein ferritin were evaluated. The first is responsible for the extrusion of iron ions from a cell and the second for the sequestration of iron ions in a non-toxic form inside the cells. The data obtained on Ferritin expression (Figure 10a,b) confirmed that, in both gastric and intestinal in vitro models, the treatments with OvT and bLact were similar or little increased compared to the control samples. In particular, the effects on ferritin expression were greater with holo-OvT 15% in GTL-16 cells, which was more evident with the 30 μg/mL concentration compared to the control (*p* < 0.05) and to the 50 μg/mL concentration (about 21%), and to both concentrations of holo-OvT 100% (about 90% to 30 μg/mL and about 64% to 50 μg/mL *p* < 0.05) and even to the two concentrations of bLact tested (about 82% to 30 μg/mL and about 29% to 50 μg/mL *p* < 0.05). Regard the Caco-2 cells (Figure 10b), holo-OvT 15% at 30 μg/mL showed a higher expression of Ferritin compared to the 50 μg/mL (about 75%) and to both concentrations of bLact (about 20% to 30 μg/mL and about 57% to 50 μg/mL *p* < 0.05), and also to holo-OvT 100% at the same concentration (about 57%, *p* < 0.05). The results allow the exclusion of iron accumulation for all tested products, thus also excluding intestinal irritability.

Finally, we analysed the expression of ferroportin because it is an important transmembrane protein known to carry iron from inside the cell to the outside. As shown in Figure 11a,b, its expression increased throughout the stimulation time (*p* < 0.05 vs. control), which indicates the working as an active extrusion mechanism. Again, on GTL-16 cells, holo-OvT 15% saturated 30 μg/mL appeared to be superior to the control (*p* < 0.05) and comparable to the highest concentration of 50 μg/mL of the tested bLact (about 1.5 times more). In particular, holo-OvT 15% saturated 30 μg/mL more effectively stimulated the ferroportin expression compared to the other tested products (about 33% with respect to holo-OvT 100% saturated 30 μg/mL and about 54% with respect to bLact and holo-OvT 15% saturated 30 μg/mL). By comparison, regarding Caco-2 cells, (Figure 11b), holo-OvT 15% at 30 μg/mL showed a similar expression of ferroportin compared to the 50 μg/mL and to both concentrations of bLact, and also to holo-OvT 100% at the same concentration. All these findings also support the hypothesis of the absence of iron accumulation in vivo and demonstrate the ability of holo-OvT 15% to carry out its effects on iron metabolism in a better manner compared to the other products tested. Overall, these experiments confirmed that holo-OvT 15% iron saturated is able to exert its positive effects on the iron metabolism, and particularly on the main proteins involved in the transport and storage of iron ions in the cells, in a better manner compared to the holo-OvT 100% saturated and tested bLact.

A recent in vitro study showed the influence of bLact on intestinal barrier function by increasing the resistance and permeability and acting on tight junctions (TJs) such as ZO-1, claudin 1 and occludin. As illustrated in Figure 12, bLact confirmed its beneficial effects on all TJ markers and the holo-OvT forms were shown to act similarly, suggesting their effective and beneficial use in humans. In particular, regarding Zo-1 (Figure 12a), a member of the junctional adhesion molecule family able to maintain the TJ structure and to modulate barrier integrity, the two holo-OvT forms (15% and 100% saturated) at both 50 and 30 μg/mL were able to increase the amount of the junction protein in a similar manner, with a greater effect at 30 μg/mL compared to the control (*p* < 0.05) and to 50 μg/mL (about 6% for 100% and about 30% for 15%). The 30 μg/mL concentration was also confirmed to be a better choice compared to bLact (about 60% to 30 μg/mL and about 25% to 50 μg/mL). Similar data were also observed regarding claudin 1 (Figure 12b), a member of the claudin family considered the principal barrier-forming protein; in particular, holo-OvT 15% 30 μg/mL was able to induce a greater effect compared to 50 μg/mL (about 15%, *p* < 0.05) and to 30 μg/mL 100% (about 10%, *p* < 0.05), and also to both 30 μg/mL bLact (about 45%, *p* < 0.05) and 50 μg/mL (about 44%, *p* < 0.05). Finally, the results obtained on occludin (Figure 12c), which contributes to TJ stabilisation and optimal barrier function, showed the best effects with holo-OvT 30 μg/mL 100% and 15% compared to all the other products tested (*p* < 0.05), also indicating the importance of the dosage to the degree of iron saturation. These effects were statistically more pronounced than bLact (*p* < 0.05) used at both concentrations. Overall, the results directly demonstrated that holo-OvT can increase TJ protein production and thus strengthen barrier function better than bLact.

### 3.5. Effects of Holo-OvT and bLact on 3D Model in Presence of Iron

Further experiments were carried out to study the biological effect of OvT and bLact mimicking the possible effects in humans due to the consumption of the test products in the presence of food (iron, 50 μM Fe^3+^). In particular, because the data obtained from the 3D model were similar to those observed on the Transwell^®^ system, the analyses were carried out using the 3D model to study the absorption of iron and the main action mechanism involved. The quantitative analysis of the basolateral environment of the gastro-intestinal barrier model confirmed a sustained slow absorption and release over time (3 and 6 h), confirming the presence of a slow-release effect (Figure 13a). In particular, at 3 h holo-OvT 15% μg/mL showed better results compared to the control (*p* < 0.05) and the tested bLact (*p* < 0.05, about 21% compared to 30 μg/mL and about 90% compared to 50 μg/mL). The analysis also confirmed that the lowest 15% holo-OvT concentration used, 30 μg/mL, had a greater effect compared to the concentration of 50 μg/mL on both times of stimulation (about 16% at 3 h and about 30% at 6 h). Furthermore, apo-OvT at concentrations of 30 and 50 μg/mL was shown to be strongly absorbed over the time period tested in the presence of iron (*p* < 0.05 vs. control, *p* < 0.05 vs. tested bLact), suggesting that apo-OvT may be an optimal candidate for human use under physiological conditions and during meals. In addition, the results of quantitative analysis of total iron (Fe^2+^ plus Fe^3+^) of the basolateral environment (Figure 13b) confirmed that both OvT and bLact were able to sequester iron and enhance its bioavailability in direct correlation with the increased absorption of iron over time (3 and 6 h). In particular, 30 μg/mL holo-OvT 15% showed better long-term absorption results (after 6 h) than the control (*p* < 0.05) and also compared to bLact (*p* < 0.05, of about 50% at 30 μg/mL and about 55% at 50 μg/mL). Furthermore, apo-OvT, especially at a concentration of 50 μg/mL, has been shown to work very well in increasing the bioavailability of iron over time compared to bLact (*p* < 0.05).

### 3.6. Intracellular Mechanisms Activated by OvT and bLact on 3D Model in Presence of Iron

To better define the mechanisms involved in both gastric and intestinal environments, different markers such as DMT1, ferritin, and ferroportin were evaluated by Western blot and densitometric analysis on a 3D system treated with 50 and 30 μg/mL of all forms of OvT and bLact after the pre-treatment with 50 μM Fe^3+^. As illustrated in Figure 14, the mechanism of action of holo-Ovt forms and bLact were also confirmed in the presence of iron; in particular, in the gastric part (Figure 14a) the effect was found to be higher with the holo-OvT 15% treatment, especially 30 μg/mL compared to the control and 50 μg/mL (*p* < 0.05, about 9.5%), and to the tested bLact for both concentrations tested (about 74% compared to 30 μg/mL and about 83% to 50 μg/mL; *p* < 0.05). In addition, the apo-OvT at the highest concentration of 50 μg/mL was shown to enhance the iron absorption in this gastric part of the model by more than the tested bLact for both concentrations tested (about 51% compared to 30 μg/mL and about 83% compared to 50 μg/mL, *p* < 0.05). In the intestinal part (Figure 14b) of this model, the DMT-1 expression was increased after all treatments, with holo-OvT 15% having the best results. In particular, the 30 μg/mL concentration had the greatest effect compared to 50 μg/mL (about 3%) and also to both concentrations of holo-OvT 100% (about 12.5% to the same concentration and about 40% to 50 μg/mL). In addition, like apo-OvT, it was also shown to be superior to bLact at both concentrations (*p* < 0.05). The iron uptake mechanism using DMT-1 expression was confirmed to be active in both sides (gastric and intestinal), confirming again that OvT and bLact share the same mechanism of iron transport.

The data obtained on Ferritin expression (Figure 15a,b) in 3D the gastro-intestinal barrier model pre-treated with Fe^3+^ (50 μM), and followed by treatments with 50 and 30 μg/mL of different OvT forms and bLact, confirmed that in both gastric and intestinal in vitro models the treatments with OvT forms and bLact were similar. In particular, the results obtained from both compartments (Figure 15) showed that there was no ferritin overexpression compared to the control. Holo-OvT 15% saturated at 30 μg/mL concentration was shown to have a comparable effect to that of its higher concentration 50 μg/mL (*p* < 0.05 vs. control). Both concentrations were shown to under-express ferritin in both models (gastric and intestinal) compared to the control (*p* < 0.05), thus excluding the risk of gastric and intestinal damage. The results of the apo-OvT at the 30 μg/mL concentration in the intestinal compartment were also shown to under-express ferritin. The results enable exclusion of iron accumulation for all tested products, thus also excluding potential intestinal damage in the presence of Fe^3+^.

Furthermore, the expression of ferroportin was analysed (Figure 16). This molecule is of great importance, because iron from foods is absorbed into the cells of the small intestine, and ferroportin allows iron to be transported out of those cells and into the bloodstream. As shown in Figure 16a,b, its expression increased throughout the stimulation time (*p* < 0.05 vs. control), which indicates the function as an active extrusion mechanism for all tested products. These results support the hypothesis of the absence of iron accumulation. Here, a greater transfer across the gastro-intestinal barrier model was again observed with 30 μg/mL holo-OvT 15% (*p* < 0.05) compared to 50 μg/mL (about 6% on gastric cells and 15% at the intestinal level) and also to bLact at 30 μg/mL (about 83% at the gastric and about 52% at the intestinal level), demonstrating its effects on iron metabolism in a better manner compared to the other products. This study also confirms that apo-OvT is comparable to bLact.

Finally, to confirm a better influence of these tested products on intestinal barrier function, the analysis of TJs such as ZO-1, claudin 1 and occludin was carried out by testing all OvT forms and bLact under 50 μM Fe^3+^ pre-treatment. As illustrated in Figure 17, bLact confirmed its effects on all TJ markers, and holo-OvT forms also demonstrated their effects to induce all TJs, consequently strengthening the integrity and the function of the intestinal barrier. In particular, regarding Zo-1 (Figure 17a), the two holo-OvT forms at both 50 and 30 μg/mL were able to increase the amount of protein in a similar manner, with a greater effect at 30 μg/mL compared to the control (*p* < 0.05) and to 50 μg/mL (*p* < 0.05 vs. 100% and about 35% for 15%). Holo-OvT 30 μg/mL was confirmed to be a better choice also with respect to bLact (about 17% compared to 30 μg/mL and about 18% compared to 50 μg/mL, *p* < 0.05). In addition, the apo-OvT forms were able to induce an increase comparable to 30 μg/mL holo-OvT. Similar data were also observed for claudin 1 (Figure 17b); in particular, holo-OvT 15% 30 μg/mL was able to induce a greater effect compared to 50 μg/mL (about 58%, *p* < 0.05) and to 30 μg/mL 100% (about 35%, *p* < 0.05), and also to both 30 μg/mL bLact (about 74%, *p* < 0.05) and 50 μg/mL (about 77%, *p* < 0.05). The 50 μg/mL concentration of apo-Ovt was confirmed to have a greater effect compared to all the products tested. Finally, the results obtained from occludin (Figure 17c) showed the main effects with holo-OvT 30 μg/mL 15% compared to all other agents tested (*p* < 0.05), indicating the importance of the dosage and the iron saturation grade. These effects were more evident compared to bLact (*p* < 0.05) used at both concentrations. Moreover, regarding occludin, 50 μg/mL apo-Ovt was confirmed to have a greater effect compared to all products tested (*p* < 0.05).

Overall, the results directly demonstrated that holo-OvT has the ability to increase TJ protein production and thereby strengthen barrier function better than bLact. The apo-OvT concentration of 50 μg/mL was demonstrated to significantly increase the expression of three tight junction proteins, even more than holo-OvT and bLact (*p* < 0.05).

## 4. Discussion

It is commonly accepted that human lactoferrin is an important component of innate immunity, and can interact with microorganisms by the iron sequestration mechanism and microbial molecules at the intestinal lumen [38]. Human lactoferrin is a glycoprotein found in milk and is also present in most exocrine secretions such as tears, saliva, intestinal mucus, and genital secretions, which have been demonstrated to have several properties including antibacterial, immuno-modulating, and anti-inflammatory properties [39,40], because the main function has a protective effect on intestinal barrier integrity [41]. The bovine lactoferrin (bLF) effects depend on its relative resistance to proteolytic digestion [21]; indeed, in vitro tests have shown that it is not totally degraded in the luminal environment of the stomach and at the level of the small intestine, allowing it to bind to specific receptors in the brush border membranes [21]. These receptors specifically mediate the uptake of bLF into enterocytes and crypt cells. In vitro, bLF is transported from the intestinal lumen to the bloodstream, and acts not only at the luminal intestinal level but also systemically [21]. Ovotransferrin, a protein abundant in hen egg white, shares many of the same activities as human/bovine lactoferrin. Ovotransferrin combines iron transport and the defence functions of mammalian serum transferrin and lactoferrin, respectively. It also shares approximately 50% of the sequence homology with each protein. However, the structural analogy between ovotransferrin and lactoferrin is much closer than the sequence homology and similar clusters of positively charged residues responsible for different activities such as antiviral/bacterial, immunomodulatory, antioxidant and anti-inflammatory properties [42]. In addition, this protein has been demonstrated to be able to supply iron to cells [43]. Ovotransferrin is readily digested by pepsin in the stomach and the bioactive forms of ovotransferrin are readily transported into human intestinal cells [42,44]. Despite a large body of evidence that supports the beneficial impact of various lactoferrin preparations on human health, there is still very little information regarding the mode of action of lactoferrin, and whether saturation of the protein with Fe affects its function and is also similarly observed with ovotransferrin. Given this premise, in this study we analysed the release, absorption, and biological activity of ovotransferrin from hen egg white with a different iron saturation rate and bovine lactoferrin using a well-characterised in vitro model of the digestive system [28], mimicking the human gastro-intestinal system both in the presence or absence of Fe^3+^. The findings obtained in this study demonstrate that holo-OvT was able to maintain a higher significant beneficial effect on cell viability and integrity compared to the other holo-OvT and bLact tested, in both gastric and intestinal compartments, confirming the important hypothesis of the iron saturation rate to obtain beneficial effects. The knowledge that the administration of ovotransferrin influences iron absorption and metabolism is a noteworthy finding. Therefore, the passage and the real absorption rate of all forms of holo-OvT mimicking oral intake, compared to bLact, confirmed a slow release. This is important data for defining the dose form and the posology in humans to prevent the rebound effect. According to the data obtained, holo-OvT promotes iron absorption while also preventing the body from being damaged by excessive iron; this condition was observed during pre-treatment with Fe^3+^. Furthermore, the concentration to obtain the beneficial effects, while simultaneously preventing the negative effects of accumulation, was shown to be 30 μg/mL of holo-OvT. Because iron metabolism is influenced by various conditions, the importance of the different test products on iron uptake (DMT1 analysis) and on the transfer across the cell monolayer (ferritin light chain for transportation and ferroportin to extrude) was investigated according to a standard protocol [28]. In addition, the tests on the molecular mechanisms demonstrate that holo-OvT can physiologically regulate iron metabolism. Indeed, it was observed that holo-OvT requires iron by transferrin, which is released as ferrous ions translocating via DMT1 into cytoplasm where it is sequestered by ferritin. The release of iron from this protein to cytoplasm occurs after reduction of ferric to ferrous ions. Then, ferrous ions are exported into plasma by ferroportin. The epithelium of the intestinal mucosa is one of the most important barriers in terms of extension that separates the intestine and the internal organs. For this reason, it is a dual target of any toxic insult from drugs or substances in the diet. In fact, it is known that the mucosal alterations cause not only damage to the tissue itself but can also represent an uncontrolled passage of potentially toxic substances from the intestinal lumen to blood [28]. In particular, as reported in the literature, bLact can have a direct effect on the intestinal epithelium and play a beneficial role on the intestinal epithelial barrier through the NK-κB signalling pathway or by preserving the integrity of the intestinal barrier [21,38]. bLF is transported from the intestinal lumen to the bloodstream and functions not only in the intestinal lumen, but also systemically [45]. The formation of tight junctions (TJs) in epithelial cells plays a pivotal role in the intestinal barrier [46]. TJs mediated by proteins such as claudins, occludin, and zonula occludens (ZO) are necessary for epithelial barrier maintenance [47,48]. For this reason, disruption of the intestinal epithelial barrier can increase intestinal permeability [49]. Therefore, the effect of bLact on the three TJs was confirmed in the present study, and the effect of OvT to significantly increasethe expression of the TJ proteins claudin-1, occludin and ZO-1 was demonstrated, effectively strengthening the integrity and the barrier function of the two cellular models used. Furthermore, the stress potentially caused by all forms of Ovt was maintained at a physiological level, thus ensuring safety without adverse or harmful effects on the intestinal epithelium, which is very important for maintaining the integrity of TJ. Therefore, the current data indicate that OvT significantly increases the expression of TJ proteins and protects the function of the intestinal epithelial barrier, and even improves the integrity of the intestinal epithelial barrier, as demonstrated by the increase in TEER values, both in the stomach and in the intestine. The same data were observed in presence of iron, and these observations indicate that OvT formulation is suitable for gastric and intestinal epithelial cells, exerting a beneficial role in terms of iron absorption and metabolism and gastro-intestinal barrier integrity.

## 5. Conclusions

In conclusion, this in vitro study demonstrates for the first time the ability of hen egg white ovotransferrin, in two different forms, the apo- (iron-free) and holo-form (containing iron at different saturation percentages), to actively enhance absorption of iron at the gastric and intestinal level without accumulation and gastric-intestinal irritability, indicating the huge potential of ovotransferrin in new dietary supplementation strategies. The demonstrated action is similar to that of bLact but with a greater effect. Major human iron markers such as DMT-1, ferritin and ferroportin were used to investigate the underlying iron uptake mechanism.

Overall, both apo-ovotransferrin and holo-ovotransferrin 15% saturated showed better performance compared to the tested bovine-lactoferrin. In detail, the 15% saturated holo-ovotransferrin showed the best activity at the lowest concentration among those used, i.e., 30 μg/mL.

Furthermore, ovotransferrin decreased the negative effects of the presence of iron, maintaining the integrity of gastro-intestinal barrier. Based on these in vitro data, we can conclude that the natural protein ovotransferrin from hen egg white is an excellent candidate for iron supplementation, as an iron absorption enhancer, and as an alternative to bovine-lactoferrin. In vivo studies should be performed to confirm the in vitro data presented in this study.

## 6. Patents

Bioseutica BV; Ovotransferrins for use in the treatment of iron deficiency anaemia. EP 21158368.7, 22 February 2021.

## Figures and Tables

**Figure 1 biomedicines-09-01543-f001:**
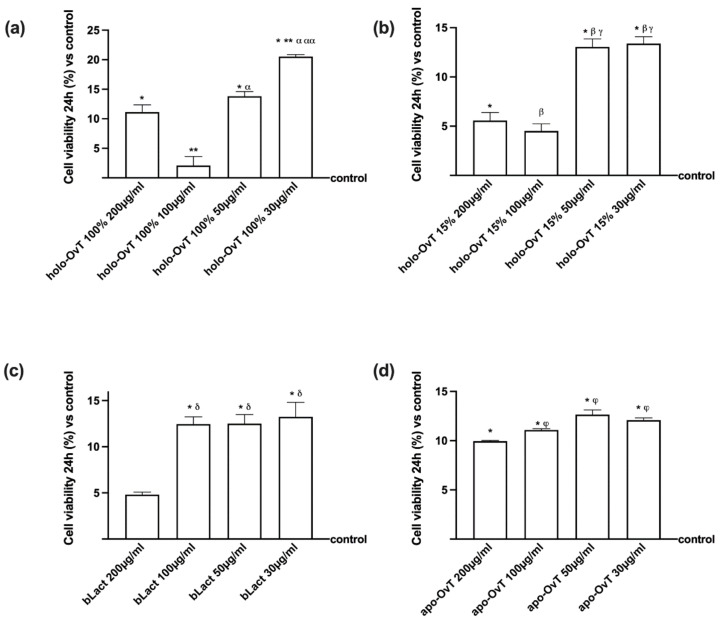
Dose-response study of cell viability on GTL-16 cells treated with different concentrations and different iron saturation types of OvT and bLact. Data are expressed as means ± SD compared to control (0% line) of four independent experiments produced in triplicate. (**a**) holo-OvT 100% saturated iron * *p* < 0.05 vs. control; ** *p* < 0.05 vs. holo-Ovt 100% 200 μg/mL; α *p* < 0.05 vs. holo-Ovt 100% 100 μg/mL; αα *p* < 0.05 vs. holo-Ovt 100% 50 μg/mL. (**b**) holo-OvT 15% saturated iron * *p* < 0.05 vs. control; β *p* < 0.05 vs. holo-Ovt 15% 200 μg/mL; γ *p* < 0.05 vs. holo-Ovt 15% 100 μg/mL. (**c**) bLact = bovine lactoferrin * *p* < 0.05 vs. control; δ *p* < 0.05 vs. bLact 200 μg/mL; (**d**) apo-Ovt * *p* < 0.05 vs. control; φ *p* < 0.05 vs. apo-Ovt 200 μg/mL.

**Figure 2 biomedicines-09-01543-f002:**
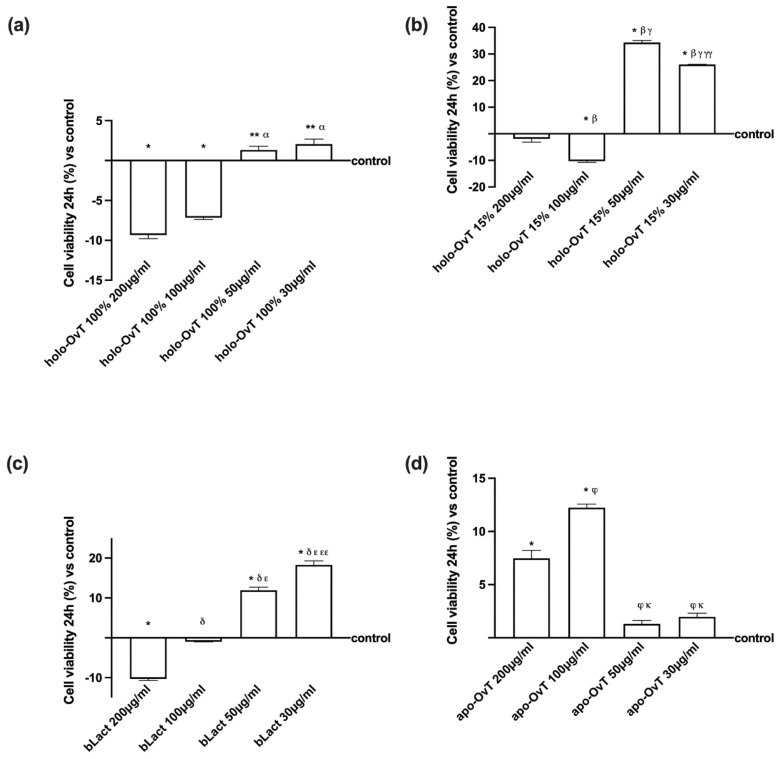
Dose-response study of cell viability on Caco-2 cell treated with different concentrations and different iron saturation types of OvT and bLact. Data are expressed as means ± SD compared to control (0% line) of four independent experiments produced in triplicate. (**a**) holo-OvT 100% saturated iron. * *p* < 0.05 vs. control; ** *p* < 0.05 vs. holo-Ovt 100% 200 μg/mL; α *p* < 0.05 vs. holo-Ovt 100% 100 μg/mL. (**b**) holo-OvT 15% saturated iron * *p* < 0.05 vs. control; β *p* < 0.05 vs. holo-Ovt 15% 200 μg/mL; γ *p* < 0.05 vs. holo-Ovt 15% 100 μg/mL; γγ *p* < 0.05 vs. holo-Ovt 15% 50 μg/mL. (**c**) bLact = bovine lactoferrin * *p* < 0.05 vs. control; δ *p* < 0.05 vs. bLact 200 μg/mL; ε *p* < 0.05 vs. bLact 100 μg/mL; εε *p* < 0.05 vs. bLact 50 μg/mL. (**d**) apo-Ovt * *p* < 0.05 vs. control; φ *p* < 0.05 vs. apo-Ovt 200 μg/mL; κ *p* < 0.05 vs. apo-Ovt 100 μg/mL.

**Figure 3 biomedicines-09-01543-f003:**
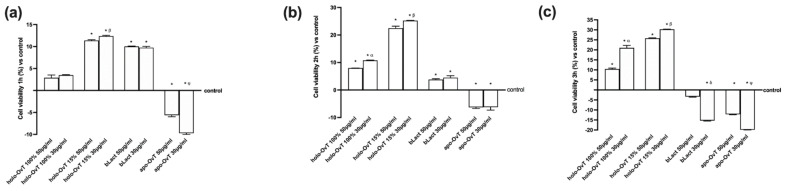
Time-course study of cell viability on GTL-16 cells treated with 50 and 30 μg/mL of OvT and bLact for 1, 2 and 3 h. (**a**) 1 h of stimulation * *p* < 0.05 vs. control; β *p* < 0.05 vs. holo-Ovt 15% 50 μg/mL; φ *p* < 0.05 vs. apo-Ovt 50 μg/mL. (**b**) 2 h of stimulation * *p* < 0.05 vs. control; α *p* < 0.05 vs. holo-Ovt 100% 50 μg/mL; β *p* < 0.05 vs. holo-Ovt 15% 50 μg/mL. (**c**) 3 h of stimulation * *p* < 0.05 vs. control; α *p* < 0.05 vs. holo-Ovt 100% 50 μg/mL; β *p* < 0.05 vs. holo-Ovt 15% 50 μg/mL; δ *p* < 0.05 vs. bLact 50 μg/mL; φ *p* < 0.05 vs. apo-Ovt 50 μg/mL. The abbreviations are the same reported in Figure 1. Data are expressed as means ± SD (%) compared to control (0% line) of four independent experiments produced in triplicate.

**Figure 4 biomedicines-09-01543-f004:**
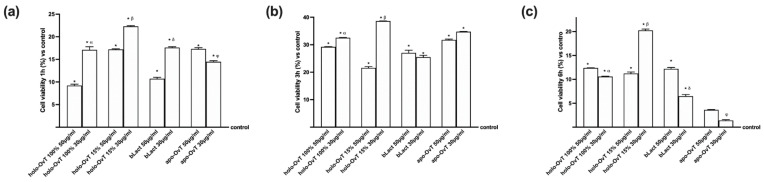
Time-course study of cell viability on Caco-2 cells treated with 50 and 30 μg/mL of OvT and bLact for 1, 3 and 6 h. (**a**) 1 h of stimulation * *p* < 0.05 vs. control; α *p* < 0.05 vs. holo-Ovt 100% 50 μg/mL; β *p* < 0.05 vs. holo-Ovt 15% 50 μg/mL; δ *p* < 0.05 vs. bLact 50 μg/mL; φ *p* < 0.05 vs. apo-Ovt 50 μg/mL. (**b**) 3 h of stimulation * *p* < 0.05 vs. control; α *p* < 0.05 vs. holo-Ovt 100% 50 μg/mL; β *p* < 0.05 vs. holo-Ovt 15% 50 μg/mL. (**c**) 6 h of stimulation * *p* < 0.05 vs. control; α *p* < 0.05 vs. holo-Ovt 100% 50 μg/mL; β *p* < 0.05 vs. holo-Ovt 15% 50 μg/mL; δ *p* < 0.05 vs. bLact 50 μg/mL; φ *p* < 0.05 vs. apo-Ovt 50 μg/mL. The abbreviations are the same reported in Figure 1. Data are expressed as means ± SD (%) compared to control (0% line) of four independent experiments produced in triplicate.

**Figure 5 biomedicines-09-01543-f005:**
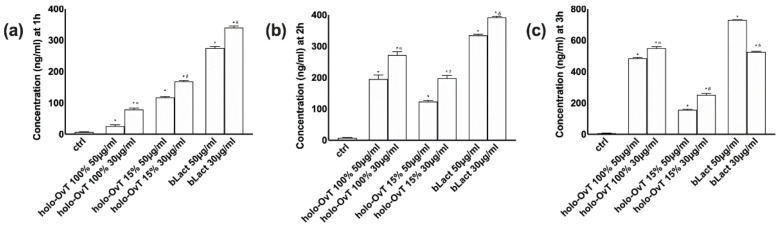
Transferrin absorption measured at the basolateral level on Transwell^®^ over time. GTL-16 cells treated with 50 and 30 μg/mL of different holo-OvT and bLact. In (**a**) the measurement at 1 h * *p* < 0.05 vs. control; α *p* < 0.05 vs. holo-Ovt 100% 50 μg/mL; β *p* < 0.05 vs. holo-Ovt 15% 50 μg/mL; δ *p* < 0.05 vs. bLact 50 μg/mL. (**b**) at 2 h * *p* < 0.05 vs. control; α *p* < 0.05 vs. holo-Ovt 100% 50 μg/mL; β *p* < 0.05 vs. holo-Ovt 15% 50 μg/mL; δ *p* < 0.05 vs. bLact 50 μg/mL. (**c**) at 3 h of stimulation * *p* < 0.05 vs. control; α *p* < 0.05 vs. holo-Ovt 100% 50 μg/mL; β *p* < 0.05 vs. holo-Ovt 15% 50 μg/mL; δ *p* < 0.05 vs. bLact 50 μg/mL. The abbreviations are the same as reported in Figure 1. Data are expressed as means ± SD (%) calculated on ng/mL compared to control (0% line) of four independent experiments produced in triplicate.

**Figure 6 biomedicines-09-01543-f006:**
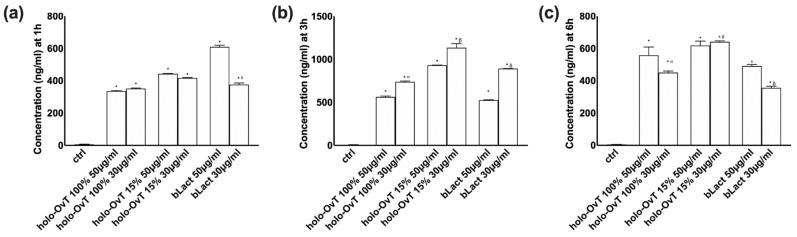
Transferrin absorption measured at the basolateral level on Transwell^®^ during time. Caco-2 cells treated with 50 and 30 μg/mL of different holo-OvT and bLact. In (**a**) the measurement at 1 h * *p* < 0.05 vs. control; δ *p* < 0.05 vs. bLact 50 μg/mL. In (**b**) at 3 h * *p* < 0.05 vs. control; α *p* < 0.05 vs. holo-Ovt 100% 50 μg/mL; β *p* < 0.05 vs. holo-Ovt 15% 50 μg/mL; δ *p* < 0.05 vs. bLact 50 μg/mL. In (**c**) at 6 h of stimulation * *p* < 0.05 vs. control; α *p* < 0.05 vs. holo-Ovt 100% 50 μg/mL; β *p* < 0.05 vs. holo-Ovt 15% 50 μg/mL; δ *p* < 0.05 vs. bLact 50 μg/mL. The abbreviations are the same as reported in Figure 1. Data are expressed as means ± SD (%) calculated on ng/mL compared to control (0% line) of four independent experiments produced in triplicate.

**Figure 7 biomedicines-09-01543-f007:**
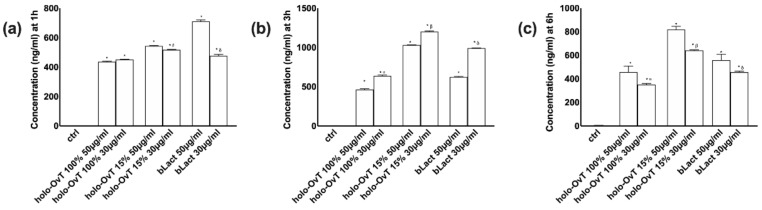
Transferrin absorption measured in a VITVO^®^ basolateral environment after Caco-2 cells. The stimulation includes 50 and 30 μg/mL of different holo-OvT and bLact. In (**a**) the measurement at 1 h * *p* < 0.05 vs. control; β *p* < 0.05 vs. holo-Ovt 15% 50 μg/mL; δ *p* < 0.05 vs. bLact 50 μg/mL. In (**b**) at 3 h * *p* < 0.05 vs. control; α *p* < 0.05 vs. holo-Ovt 100% 50 μg/mL; β *p* < 0.05 vs. holo-Ovt 15% 50 μg/mL; δ *p* < 0.05 vs. bLact 50 μg/mL. In (**c**) at 6 h of stimulation * *p* < 0.05 vs. control; α *p* < 0.05 vs. holo-Ovt 100% 50 μg/mL; β *p* < 0.05 vs. holo-Ovt 15% 50 μg/mL; δ *p* < 0.05 vs. bLact 50 μg/mL. The abbreviations are the same as reported in Figure 1. Data are expressed as means ± SD (%) calculated on ng/mL compared to control (0% line) of four independent experiments produced in triplicate.

**Figure 8 biomedicines-09-01543-f008:**
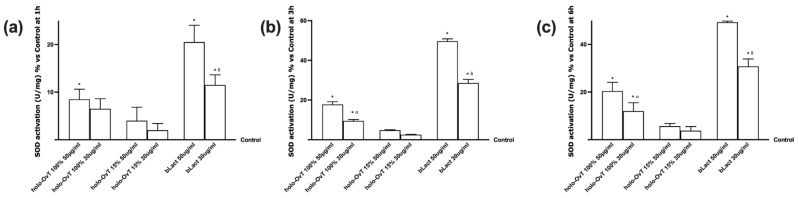
SOD activity measured on VITVO^®^ after Caco-2 cells. The stimulation includes 50 and 30 μg/mL of different holo-OvT and bLact. In (**a**) the measurement at 1 h * *p* < 0.05 vs. control; δ *p* < 0.05 vs. bLact 50 μg/mL. In (**b**) at 3 h * *p* < 0.05 vs. control; α *p* < 0.05 vs. holo-Ovt 100% 50 μg/mL; δ *p* < 0.05 vs. bLact 50 μg/mL. In (**c**) at 6 h of stimulation * *p* < 0.05 vs. control; α *p* < 0.05 vs. holo-Ovt 100% 50 μg/mL; δ *p* < 0.05 vs. bLact 50 μg/mL. The abbreviations are the same as reported in Figure 1. Data are expressed as means ± SD (%) calculated on U/mg compared to control (0% line) of four independent experiments produced in triplicate.

**Figure 9 biomedicines-09-01543-f009:**
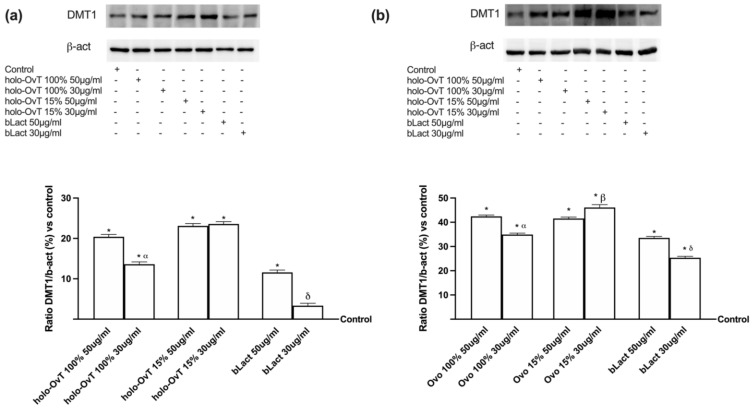
Western blot and densitometric analysis of DMT-1 on total lysates of GTL-16 (**a**) and Caco-2 cells (**b**) of the VITVO^®^ system. The cells were treated with 50 and 30 μg/mL of different OvT and bLact. The images shown are an example of each protein of four independent experiments reproduced in triplicate. The abbreviations are the same as reported in Figure 1. Data are expressed as means ± SD (%) of four independent experiments normalised and verified on β-actin detection. In (**a**) * *p* < 0.05 vs. control; α *p* < 0.05 vs. holo-Ovt 100% 50 μg/mL; δ *p* < 0.05 vs. bLact 50 μg/mL. In (**b**) * *p* < 0.05 vs. control; α *p* < 0.05 vs. holo-Ovt 100% 50 μg/mL; β *p* < 0.05 vs. holo-Ovt 15% 50 μg/mL; δ *p* < 0.05 vs. bLact 50 μg/mL.

**Figure 10 biomedicines-09-01543-f010:**
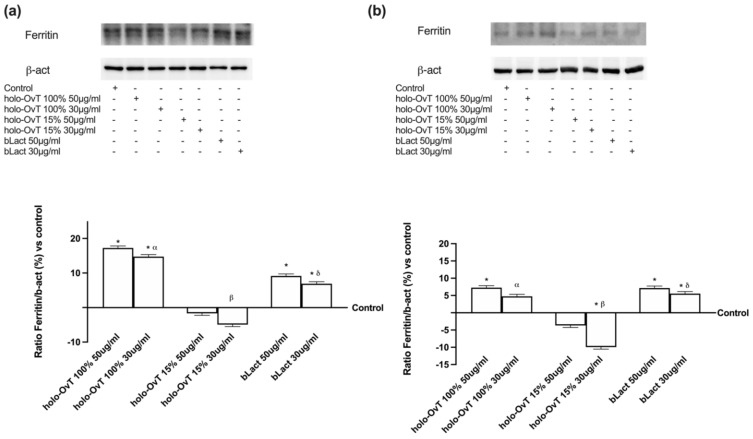
Western blot and densitometric analysis of Ferritin on total lysates of GTL-16 (**a**) and Caco-2 cells (**b**) of the VITVO^®^ system. The cells were treated with 50 and 30 μg/mL of different OvT and bLact. The images shown are an example of each protein of four independent experiments reproduced in triplicates. The abbreviations are the same as reported in Figure 1. Data are expressed as means ± SD (%) of four independent experiments normalised and verified on β-actin detection. In (**a**) * *p* < 0.05 vs. control; α *p* < 0.05 vs. holo-Ovt 100% 50 μg/mL; β *p* < 0.05 vs. holo-Ovt 15% 50 μg/mL; δ *p* < 0.05 vs. bLact 50 μg/mL. In (**b**) * *p* < 0.05 vs. control; α *p* < 0.05 vs. holo-Ovt 100% 50 μg/mL; β *p* < 0.05 vs. holo-Ovt 15% 50 μg/mL; δ *p* < 0.05 vs. bLact 50 μg/mL.

**Figure 11 biomedicines-09-01543-f011:**
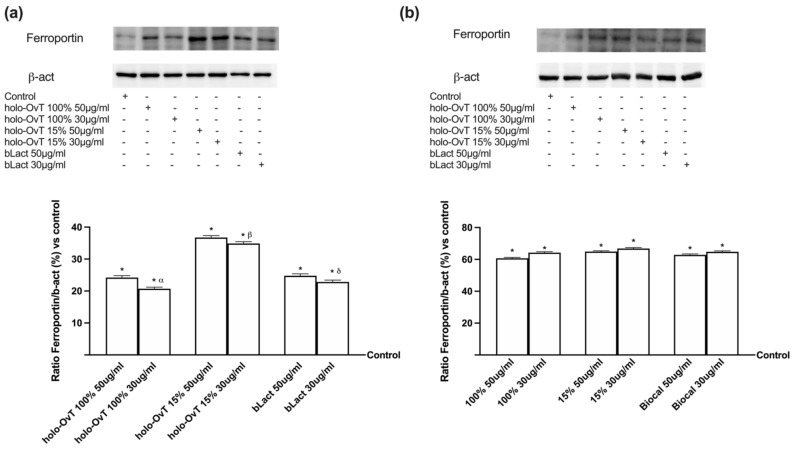
Western blot and densitometric analysis of ferroportin on total lysates of GTL-16 (**a**) and Caco-2 cells (**b**) of the VITVO^®^ system. The cells were treated with 50 and 30 μg/mL of different OvT and bLact. The images shown are an example of each protein of four independent experiments reproduced in triplicate. The abbreviations are the same as reported in Figure 1. Data are expressed as means ± SD (%) of four independent experiments normalised and verified on β-actin detection. In (**a**) * *p* < 0.05 vs. control; α *p* < 0.05 vs. holo-Ovt 100% 50 μg/mL; β *p* < 0.05 vs. holo-Ovt 15% 50 μg/mL; δ *p* < 0.05 vs. bLact 50 μg/mL. In (**b**) * *p* < 0.05 vs. control.

**Figure 12 biomedicines-09-01543-f012:**
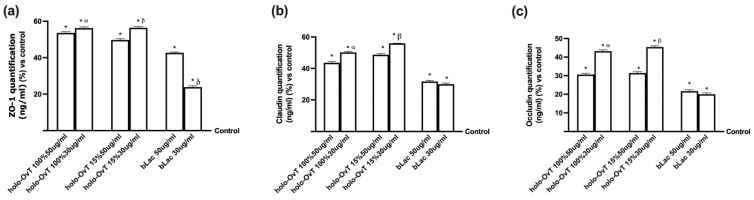
Analysis of TJ on Caco-2 lysates. (**a**) Zo-1; (**b**) claudin; (**c**) occludin measured by ELISA test. The cells were treated with 50 and 30 μg/mL of different OvT and bLact. The abbreviations are the same as reported in Figure 1. Data are expressed as means ± SD (%) of four independent experiments normalised to control the sample (0% line). In (**a**) * *p* < 0.05 vs. control; α *p* < 0.05 vs. holo-Ovt 100% 50 μg/mL; β *p* < 0.05 vs. holo-Ovt 15% 50 μg/mL; δ *p* < 0.05 vs. bLact 50 μg/mL. In (**b**) * *p* < 0.05 vs. control; α *p* < 0.05 vs. holo-Ovt 100% 50 μg/mL; β *p* < 0.05 vs. holo-Ovt 15% 50 μg/mL. In (**c**) * *p* < 0.05 vs. control; α *p* < 0.05 vs. holo-Ovt 100% 50 μg/mL; β *p* < 0.05 vs. holo-Ovt 15% 50 μg/mL.

**Figure 13 biomedicines-09-01543-f013:**
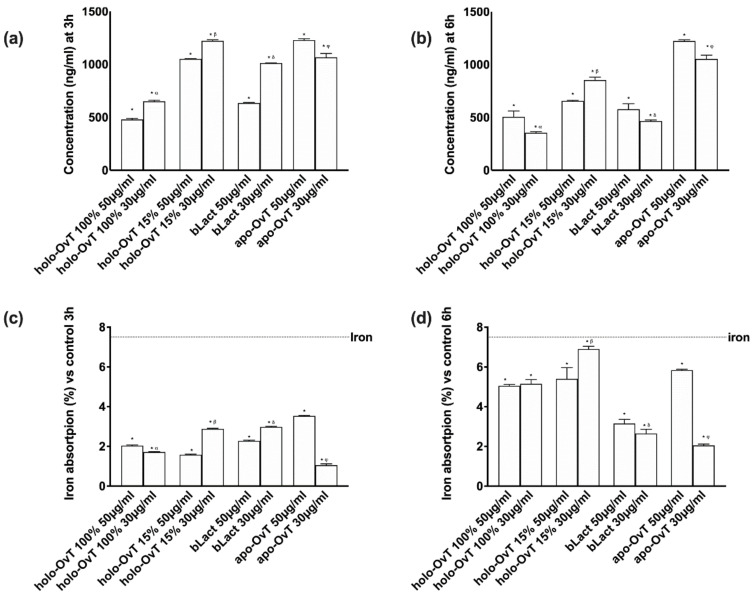
Effects of Ovt and bLact on transferrin (**a**,**b**) and iron (**c**,**d**) absorption in cells pre-treated with 50 μMFe^3+^ over time (3 and 6 h). The abbreviations are the same as reported in Figure 1. Iron = pre-stimulation with 50 μM Fe^3+^. Data are expressed as means ± SD (%) compared to control of four independent experiments produced in triplicate. In (**a**–**c**) * *p* < 0.05 vs. control; α *p* < 0.05 vs. holo-Ovt 100% 50 μg/mL; β *p* < 0.05 vs. holo-Ovt 15% 50 μg/mL; δ *p* < 0.05 vs. bLact 50 μg/mL; φ *p* < 0.05 vs. apo-Ovt 50 μg/mL. In (**d**) * *p* < 0.05 vs. control; β *p* < 0.05 vs. holo-Ovt 15% 50 μg/mL; δ *p* < 0.05 vs. bLact 50 μg/mL; φ *p* < 0.05 vs. apo-Ovt 50 μg/mL.

**Figure 14 biomedicines-09-01543-f014:**
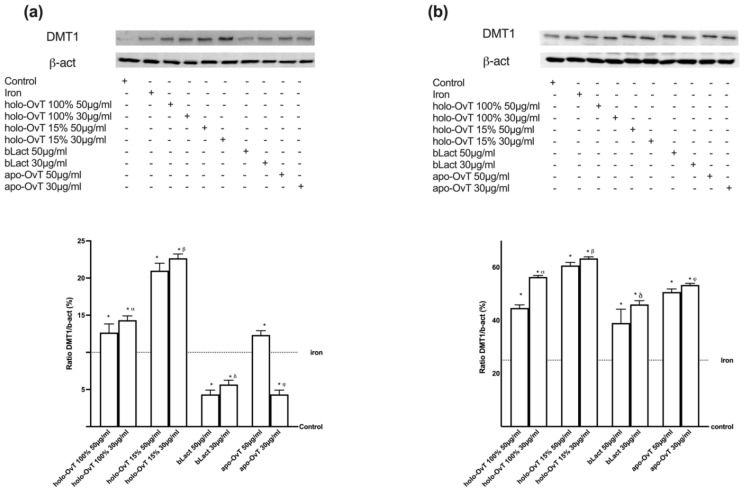
Western blot and densitometric analysis of DMT1 on total lysates of GTL-16 (**a**) and Caco-2 cells (**b**) of the VITVO^®^ system. The cells were treated with 50 and 30 μg/mL of different OvT and bLact. The images shown are an example of each protein of four independent experiments reproduced in triplicate. Iron= 50 μM Fe^3+^ and the abbreviations are the same as reported in Figure 1. Data are expressed as means ± SD (%) of four independent experiments normalised and verified on β-actin detection. In (**a**) * *p* < 0.05 vs. control; α *p* < 0.05 vs. holo-Ovt 100% 50 μg/mL; β *p* < 0.05 vs. holo-Ovt 15% 50 μg/mL; δ *p* < 0.05 vs. bLact 50 μg/mL; φ *p* < 0.05 vs. apo-Ovt 50 μg/mL. In (**b**) * *p* < 0.05 vs. control; α *p* < 0.05 vs. holo-Ovt 100% 50 μg/mL; β *p* < 0.05 vs. holo-Ovt 15% 50 μg/mL; δ *p* < 0.05 vs. bLact 50 μg/mL; φ *p* < 0.05 vs. apo-Ovt 50 μg/mL.

**Figure 15 biomedicines-09-01543-f015:**
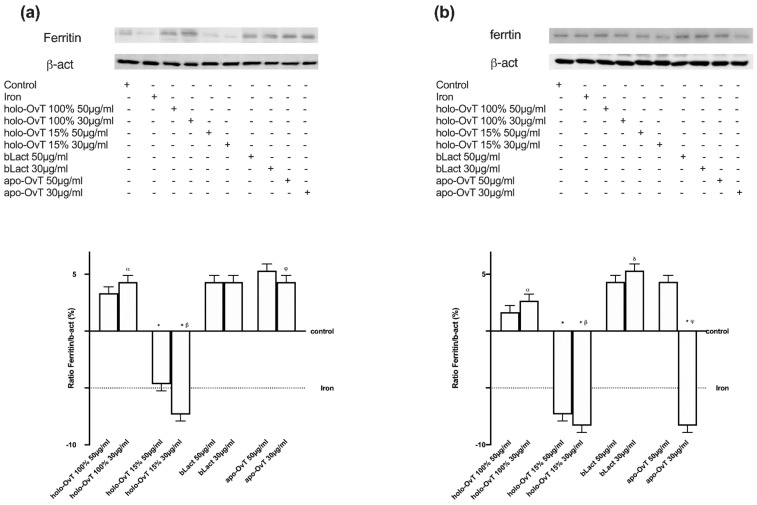
Western blot and densitometric analysis of ferritin on total lysates of GTL-16 (**a**) and Caco-2 cells (**b**) of the VITVO^®^ system. The cells were treated with 50 and 30 μg/mL of different OvT and bLact. The images shown are an example of each protein of four independent experiments reproduced in triplicate. Iron= 50 μM Fe^3+^ and the abbreviations are the same as reported in Figure 1. Data are expressed as means ± SD (%) of four independent experiments normalised and verified on β-actin detection. In (**a**) * *p* < 0.05 vs. control; α *p* < 0.05 vs. holo-Ovt 100% 50 μg/mL; β *p* < 0.05 vs. holo-Ovt 15% 50 μg/mL; φ *p* < 0.05 vs. apo-Ovt 50 μg/mL. In (**b**) * *p* < 0.05 vs. control; α *p* < 0.05 vs. holo-Ovt 100% 50 μg/mL; β *p* < 0.05 vs. holo-Ovt 15% 50 μg/mL; δ *p* < 0.05 vs. bLact 50 μg/mL; φ *p* < 0.05 vs. apo-Ovt 50 μg/mL.

**Figure 16 biomedicines-09-01543-f016:**
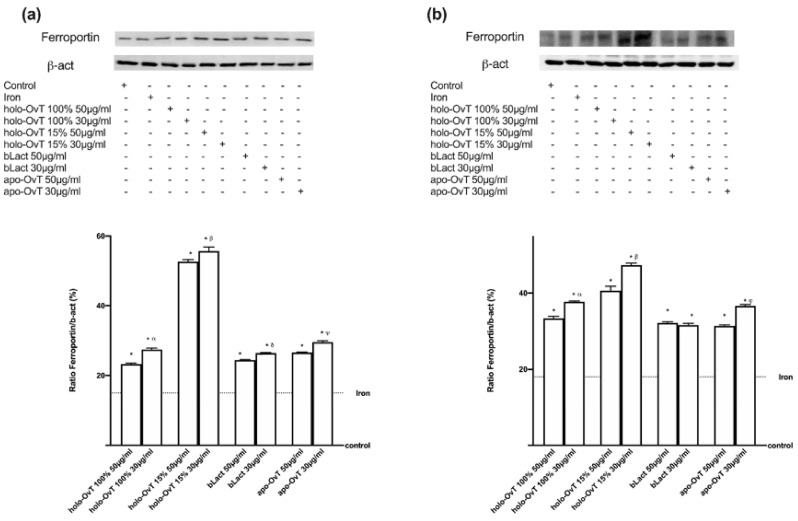
Western blot and densitometric analysis of ferroportin on total lysates of GTL-16 (**a**) and Caco-2 cells (**b**) of the VITVO^®^ system. The cells were treated with 50 and 30 μg/mL of different OvT and bLact. The images shown are an example of each protein of four independent experiments reproduced in triplicate. Iron= 50 μM Fe^3+^ and the abbreviations are the same as reported in Figure 1. Data are expressed as means ± SD (%) of four independent experiments normalised and verified on β-actin detection. In (**a**) * *p* < 0.05 vs. control; α *p* < 0.05 vs. holo-Ovt 100% 50 μg/mL; β *p* < 0.05 vs. holo-Ovt 15% 50 μg/mL; δ *p* < 0.05 vs. bLact 50 μg/mL; φ *p* < 0.05 vs. apo-Ovt 50 μg/mL. In (**b**) * *p* < 0.05 vs. control; α *p* < 0.05 vs. holo-Ovt 100% 50 μg/mL; β *p* < 0.05 vs. holo-Ovt 15% 50 μg/mL; φ *p* < 0.05 vs. apo-Ovt 50 μg/mL.

**Figure 17 biomedicines-09-01543-f017:**
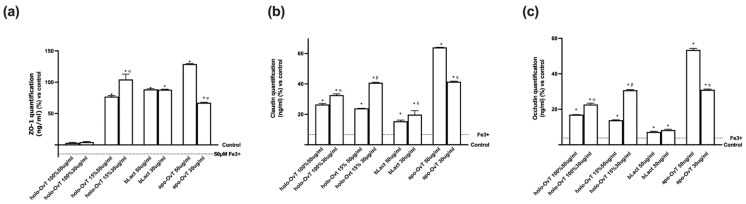
Analysis of TJ on Caco-2 lysates. (**a**) Zo-1; (**b**) claudin; (**c**) occludin measured by ELISA test. The cells were treated with 50 and 30 μg/mL of different OvT and bLact. The abbreviations are the same as reported in Figure 1. Data are expressed as means ± SD (%) of four independent experiments normalised to control sample (0% line). In (**a**) * *p* < 0.05 vs. control; α *p* < 0.05 vs. holo-Ovt 100% 50 μg/mL; φ *p* < 0.05 vs. apo-Ovt 50 μg/mL. In (**b**) * *p* < 0.05 vs. control; α *p* < 0.05 vs. holo-Ovt 100% 50 μg/mL; β *p* < 0.05 vs. holo-Ovt 15% 50 μg/mL; δ *p* < 0.05 vs. bLact 50 μg/mL; φ *p* < 0.05 vs. apo-Ovt 50 μg/mL. In (**c**) * *p* < 0.05 vs. control; α *p* < 0.05 vs. holo-Ovt 100% 50 μg/mL; β *p* < 0.05 vs. holo-Ovt 15% 50 μg/mL; φ *p* < 0.05 vs. apo-Ovt 50 μg/mL.

## Data Availability

The data presented in this study are available on reasonable request from the corresponding author. The data are not publicly available due to the presence of patent.

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
