# Peer review of "Ovotransferrin Supplementation Improves the Iron Absorption: An In Vitro Gastro-Intestinal Model"

_biomedicines, 2021, doi:10.3390/biomedicines9111543_

Round 1

Reviewer 1 Report

  1. Please provide in detail, condition used in MTT assay for control cells.
  2. Please use different statistical significance symbol, when comparing different treatment condition.
  3. Please provide better images for figure 9b, 10b, 11b, 15a, and 16b. 

Author Response

Reviewer 1

1.Please provide in detail, condition used in MTT assay for control cells.

Thank you for the comment. We have added the information required in Method section. All changes are reported in red.

2. Please use different statistical significance symbol, when comparing different treatment condition.

Thank you for the comment and we apologize for inconvenience. We have changed the symbol that indicates significance between different treatment condition. We hope that now the images are sufficiently improved.

3.Please provide better images for figure 9b, 10b, 11b, 15a, and 16b. 

Thank you for the comment. We have exported in a better quality (600dpi) the images indicated by your suggestions and we hope that they have a satisfactory resolution for publication. However, we didn’t change the background showed in the lane, in order to maintain the original version of antibody staining. Unfortunately, the contrast between the lane and the background is not very high. However, we prefer to maintain the original aspect of the image, without filters and adjustments.

In addition, we have carefully revised the manuscript to edit the typos (changes are reported in red)

Thank you for all useful suggestions and we hope that now the manuscript will be suitable for publication on “Biomedicines” journal as open access paper.

All authors have approved the new version of manuscript for submission.

Thanking you for your attention,
Sincerely yours,

                                                       Francesca Uberti

Reviewer 2 Report

The presented work has a high scientific value. It deals with the metabolism of iron, especially its resorption. It compares the effect of transferrin and ovotransferrin in vitro. The introduction is very well done, the methods used are appropriate, the results are relevant and presented in clear graphs. The conclusion is brief and concise. The work proves that ovotransferrin may be an excellent candidate for iron supplementation in humans. The bibliography is satisfactory.

Author Response

Reviewer 2

The presented work has a high scientific value. It deals with the metabolism of iron, especially its resorption. It compares the effect of transferrin and ovotransferrin in vitro. The introduction is very well done, the methods used are appropriate, the results are relevant and presented in clear graphs. The conclusion is brief and concise. The work proves that ovotransferrin may be an excellent candidate for iron supplementation in humans. The bibliography is satisfactory.

 Thank you for the comment and we are very happy to see your positive opinion about the paper.

Anyway, we have carefully revised the manuscript to edit any typos and any changes are reported in red.

Thank you for all useful suggestions and we hope that now the manuscript will be suitable for publication on “Biomedicines” journal as open access paper.

All authors have approved the new version of manuscript for submission.

Thanking you for your attention,
Sincerely yours,

                                                                                   Francesca Uberti